# Male Differentiation in the Marine Copepod *Oithona nana* Reveals the Development of a New Nervous Ganglion and Lin12-Notch-Repeat Protein-Associated Proteolysis

**DOI:** 10.3390/biology10070657

**Published:** 2021-07-13

**Authors:** Kevin Sugier, Romuald Laso-Jadart, Benoît Vacherie, Jos Käfer, Laurie Bertrand, Karine Labadie, Nathalie Martins, Céline Orvain, Emmanuelle Petit, Patrick Wincker, Jean-Louis Jamet, Adriana Alberti, Mohammed-Amin Madoui

**Affiliations:** 1Génomique Métabolique, Genoscope, Institut François Jacob, Univ Evry, Université Paris-Saclay, 91000 Evry, France; kevin.sugier@inrae.fr (K.S.); rlasojad@genoscope.cns.fr (R.L.-J.); lbertran@genoscope.cns.fr (L.B.); nmartins@genoscope.cns.fr (N.M.); corvain@genoscope.cns.fr (C.O.); pwincker@genoscope.cns.fr (P.W.); adriana.alberti-thominiaux@cea.fr (A.A.); 2Genoscope, Institut de Biologie François-Jacob, Commissariat à l’Energie Atomique (CEA), Université Paris-Saclay, 91000 Evry, France; bvacheri@genoscope.cns.fr (B.V.); klabadie@genoscope.cns.fr (K.L.); mpetit@genoscope.cns.fr (E.P.); 3Laboratoire de Biométrie et Biologie Evolutive UMR 5558, Université Lyon 1, Université de Lyon, 69622 Villeurbanne, France; jos.kafer@univ-lyon1.fr; 4Mediterranean Institute of Oceanography, Université de Toulon, Aix-Marseille Université, CEDEX 9, 83041 Toulon, France; jean-louis.jamet@univ-tln.fr

**Keywords:** copepods, sexual differentiation, nervous system, Lin12-Notch-Repeat, Oithona, protein–protein interaction

## Abstract

**Simple Summary:**

Copepods are tiny crustaceans and the most abundant animals on Earth; they also play a crucial role in the marine food chain. Among copepods, Oithona is one of most ecologically successful and is known for its differential behavior between males and females. The males adopt the strategy “live fast, die young”: they are constantly in motion to search for females, more vulnerable to predators, feed less, and have a higher mortality rate. In our study, we found the presence of a new male-specific ganglion in *Oithona nana* probably involved in female cues sensing. We also demonstrate the potential role of new Lin-12 Notch Repeat proteins in the development of this ganglion by interacting with proteins involved in the development of the nervous system. Thanks to our findings, we propose that the “live fast, die young” strategy of the *O. nana* males is optimized by the explosion of these Lin-12 Notch Repeat proteins in the male proteome involved in the development of the male-specific olfactory ganglion to increase female cue sensing and mating.

**Abstract:**

Copepods are among the most numerous animals, and they play an essential role in the marine trophic web and biogeochemical cycles. The genus Oithona is described as having the highest density of copepods. The Oithona male paradox describes the activity states of males, which are obliged to alternate between immobile and mobile phases for ambush feeding and mate searching, respectively, while the female is less mobile and feeds less. To characterize the molecular basis of this sexual dimorphism, we combined immunofluorescence, genomics, transcriptomics, and protein–protein interaction approaches and revealed the presence of a male-specific nervous ganglion. Transcriptomic analysis showed male-specific enrichment for nervous system development-related transcripts. Twenty-seven Lin12-Notch Repeat domain-containing protein coding genes (LDPGs) of the 75 LDPGs identified in the genome were specifically expressed in males. Furthermore, some LDPGs coded for proteins with predicted proteolytic activity, and proteases-associated transcripts showed a male-specific enrichment. Using yeast double–hybrid assays, we constructed a protein–protein interaction network involving two LDPs with proteases, extracellular matrix proteins, and neurogenesis-related proteins. We also hypothesized possible roles of the LDPGs in the development of the lateral ganglia through helping in extracellular matrix lysis, neurites growth guidance, and synapses genesis.

## 1. Introduction

Copepods are small planktonic crustaceans that represent the most abundant metazoan subclass on Earth, and they occupy all ecological aquatic niches [1,2]. Among them, the genus Oithona is described as having the highest numerical density [3], being the most cosmopolitan [4] and playing a key role as a secondary producer in the marine food web and in biogeochemical cycles [5]. Due to its importance and abundance, Oithona phylogeography, ecology, behavior, life cycle, anatomy, and genomics are well studied [6,7,8,9,10,11,12,13].

*Oithona* spp. are active ambush-feeding omnivores; that is, to feed, the individuals remain static, jump on prey that come within their range, and capture them with their buccal appendages [8]. While females spend the majority of their time feeding, and are thus mostly immobile, males actively seek females for mating. The mating success of males thus increases if they are motile and not feeding. Theoretically, to maximize mating success, males have to alternate between feeding and female-searching periods, which constitutes a paradox in the Oithona male behavior [8].

A multi-year survey in Little Bay of Toulon, where *O. nana* is the dominant zooplankton throughout the year with no significant seasonal variation, suggested a continuous reproduction [14] as observed in other Oithona populations [15]. Under laboratory conditions with the two sexes incubated separately, *O. nana* males have a mean lifetime of 25 days, and females live for approximately 42 days. However, the in situ lifespan is unknown, as is the reproduction rate. Nonetheless, in the case of female saturation, *O. davisae* males have a reproduction rate of 0.9 females ${\mathrm{male}}^{-1}$ ${\mathrm{day}}^{-1}$, depending on the production of spermatophores that are transferred during mating [8]. A strongly biased sex-ratio toward females (male/female ratio <0.22) was observed in the *O. nana* population of Toulon Little Bay (France) [14]. Several causes could explain this observation; one possibility is the higher male exposure to predators due to its higher motility, considered as a “risky” behavior [16,17]. However, other possibilities can be proposed, such as environmental sex determination (ESD), which has been observed in other copepods [18], or energy resource depletion as a consequence of male energy consumption during the search for mates.

Recently, the *O. nana* genome was sequenced, and its comparison to other genomes showed an explosion of Lin12-Notch-Repeat (LNR) domain-containing protein-coding genes (LDPGs) [9]. Among the LDPGs present in the genome, five were found under natural selection in Mediterranean Sea populations, including notably one point mutation generating an amino acid change within the LNR domain of a male-specific protein [9,19]. This provided the first evidence of O. nana molecular differences between sexes at the transcriptional level, and a potential new repertoire of candidate genes for functional analysis.

To further investigate the molecular basis of *O. nana* sexual differentiation, we used in this study a multi-approach analysis, including (i) in situ sex ratio determination over a fifteen-year time series, (ii) sexual system determination by sex-specific polymorphism analysis, (iii) immunofluorescence staining of the nervous system in male and female *O. nana* copepods, (iv) in silico analysis of the structure and evolution of the LDPGs, (v) sex-specific gene expression through RNA-seq analysis, and (vi) the identification of an LDP protein–protein interaction network using a yeast two-hybrid system.

## 2. Materials and Methods

### 2.1. Sex Ratio in the Toulon Little Bay

*Oithona nana* specimens were sampled on the East side of the Toulon Little Bay, France (Lat. 43°$06^{\prime }$$52.1^{\prime \prime }$ N, Long. 05°$55^{\prime }$$42.7^{\prime \prime }$ E), which is not a protected area and does not require any permission to catch plankton according to local laws and regulations. The samples were collected from the upper water layer (0–10 m) using zooplankton nets with a mesh of 90 and 200 μm. Samples were preserved in 5% formaldehyde. The monitoring of *O. nana* in Toulon Little Bay was performed monthly from 2002 to 2017. Individuals of both sexes were identified and counted under a stereomicroscope.

### 2.2. Immunofluorescence Analysis

Adult individuals of both sexes collected in June 2017 were isolated under a stereomicroscope, washed in 0.5X PBST, and incubated for 72 h at room temperature with rabbit β3-tubulin antibody (ab179513, 1:100) and 100 μL normal donkey serum. After washing with PBST, the individuals were incubated for 72 h with cy3 goat anti-rabbit conjugated IgG secondary antibody (AB2338000, 1:200, Jackson ImmunoResearch Laboratory) and normal donkey serum. Individuals were then washed with PBST and mounted on slides and microslides with Citifluor mounting solution (AF87). A similar protocol was used to label other individuals for α-tubulin using cy5-conjugated rabbit α-tubulin antibody (ab52866, 1:100). Observations were made on an Olympus BX43 fluorescence microscope, and images were taken with Toupview, then labeled with Inkscape 0.92.4.

### 2.3. Biological Materials and Rna-Seq Experiments

Plankton sampled in November 2015 and November 2016 in the Little Bay of Toulon, France were preserved in 70% ethanol and stored at −20 °C. The copepods were isolated under the stereomicroscope as previously described. We selected *O. nana* individuals from five different development stages: five pairs of egg sacs, four nauplii (larvae), four copepodites (juveniles), four adult females, and four adult males. All individuals were isolated from the November 2015 sampling, except for the eggs. Each individual was isolated, then crushed with a tissue grinder (Axygen, Corning, Arizona, MA, USA) into a 1.5 mL Eppendorf tube. Total mRNAs were extracted with the NucleoSpin RNA XS kit (Macherey-Nagel, GmbH & Co., Duren, Germany) following the manufacturer’s instructions and then quantified on a Qubit 2.0 with the RNA HS Assay kit (ThermoFisher Scientific, Waltham, MA, USA); quality was assessed on a Bioanalyzer 2100 with the RNA 6000 Pico Assay kit (Agilent, Gilent Techlonoly, Santa Clara, CA, USA). cDNAs were constructed using the SMARTer v4 Ultra low Input RNA kit (Takara, San Jose, CA, USA). After cDNA shearing using a Covaris E210 instrument, Illumina libraries were constructed using the NEBNext Ultra II kit (New England Biolabs, MA, USA) and sequenced on an Illumina HiSeq2500. A minimum of 9.7e6 reads pairs were produced from each individual (Appendix A).

### 2.4. Sex-Determination System Identification by Rna-Seq

RNA-seq reads from both sexes (four females and four males) were aligned against *O. nana* genes. Reads having an alignment length lower than 80% and nucleic identity lower than 97% were removed. The variant calling step was performed with the “samtools mpileup” and “bcftools call” commands with default parameters [20], and only bi-allelic sites were kept.

To identify the most likely sexual system in *O. nana*, we used SD-pop [21]. Just like its predecessor SEX-DETector [22], SD-pop calculates the likelihood of three sexual models (the absence of sex chromosomes, the XY system, or the ZW system), which can be compared using the Bayesian information criterion. SD-pop is based on population genetics (i.e., Hardy–Weinberg equilibrium for autosomal genes and different equilibria for sex-linked genes) instead of Mendelian transmission from parents to offspring and thus can be used without the requirement of a controlled cross.

The number of individuals used (four for each sex) is close to the lower limit for SD-pop, where the robustness of the method is weakening. To test whether the model preferred by SD-pop could have been preferred purely by chance, we permuted the sex of the individuals, with the constraint of keeping four females and four males ((8!/(4!×4!)−1= 69 permuted datasets). As the XY model is strictly equivalent to the ZW model with the sexes of all individuals changed, two SD-pop models (no sex chromosomes and ZW) were run on all possible permutations of the data, and the BIC of each model was calculated. The genes inferred as sex-linked based on their posterior probability (>0.8) were manually annotated.

### 2.5. Copepod Phylogenetic Tree

The ribosomal 18S sequences from seven arthropods, including five copepods (*O. nana*, *Lepeophtheirus salmonis*, *Tigriopus californicus*, *Eurytemora affinis*, *Calanus glacialis*, *Daphnia pulex*, and *Drosophila melanogaster*), were downloaded from NCBI. The sequences were aligned with MAFFT [23] using default parameters. The nucleotide blocks conserved among the seven species were selected by Gblock [24] on Seaview [25] and manually curated. The maximum-likelihood phylogenetic tree was constructed using PhyML 3.0 with the General Time Reversible (GTR) model and branch supports computed by the approximate likelihood ratio test (aLRT) [26].

### 2.6. Gene Annotation

The functional annotation of genes was updated from the previous genome annotation [9] using InterProScan v5.8-49.0 [27], BlastKOALA v2.1 [28], and by alignment on the NCBI non-redundant protein database using Diamond [29]. Furthermore, a list of *O. nana* genes under natural selection in the Mediterranean Sea was added based on previous population genomic analyses [19]. We further considered the annotation provided by either (i) Pfam [30] or SMART [31] protein domains, (ii) GO terms (molecular function, biological process, or cellular component) [32], (iii) KEGG pathways [33], and (iv) the presence of loci under natural selection. These four gene features were used to identify specific enrichment in a given set of genes using a hypergeometric test that estimated the significance of the intersection between a specific gene list and one of the four global annotation lists.

### 2.7. Hmm Search for Ldpgs Identification

From the InterProScan annotation of the *O. nana* proteome, 25 LNR domain sequences were detected ($p\leq 10^{-6}$), extracted, and aligned with MAFFT using default parameters. A Hidden Markov Model (HMM) was generated from the aligned sequences using the “hmmbuild” function of the HMMER tool version 3.1b1 [34]. The *O. nana* proteome was scanned by “hmmsearch” using the LNR HMM profile. Detected domains were considered canonical LNR domains for E-value, c-E-value, and i-E-value $\leq 10^{-6}$ and containing at least six cysteines or considered as LNR-like domains for E-value, c-E-value, and i-E-value between $10^{-6}$ and $10^{-1}$ and containing at least four cysteines. A weblogo [35] was generated to represent conserved residues for three LNRs of the Notch protein and the LNRs and LNR-like proteins detected by HMM. The LNR and LNR-like domain-containing proteins constituting the final LDP set was used for further analysis. Deep-Loc [36] was used to determinate the cellular localization of LDPs. To detect signal peptides and membrane protein topology, we used the online services of SignalP 5.0 [37] and TOPCONS [38], respectively.

### 2.8. Phylogeny Tree of Oithona nana Lnr Domains

The *O. nana* nucleotide sequences of the LNR and LNR-like domains were aligned using MAFFT with default parameters. The maximum-likelihood phylogenetic tree was constructed using PhyML.3.0 with a model designed by the online execution of Smart Model Selection v.1.8.1 [39], and with branch supports computed by the aLRT method. The GTR model was used with an estimated discrete gamma distribution (a = 1.418) and a proportion of fixed invariable sites (I = 0.25). The tree was visualized using MEGA-X [40].

### 2.9. Differential Expression Analysis

RNA-seq reads from 20 libraries were mapped independently against the *O. nana* genes using “bwa-mem” (v.0.7.15-r1140) with default parameters [41]. Read counts were extracted from the 20 BAM files with samtools (v.1.4) [20]. Each set of reads was validated by a pairwise MA-plot to ensure a global representation of the *O. nana* transcriptome in each sample (Appendix A). One nauplius sample showing a biased read count distribution was discarded. Read counts from valid replicates were used as input data for the DESeq R package [42] to identify differential gene expression between the five development stages through pairwise comparisons of each developmental stage. Genes having a Benjamini–Hochberg-corrected *p*-value ≤ 0.05 in one of the pairwise comparisons were considered significantly differentially expressed. To identify stage-specific genes, we selected those that were at least twice as highly expressed based on the normalized read count mean (log2(foldChange) > 1) in one development stage compared to the four others. Upregulated stage-specific genes were represented by a heatmap. The same method was used to determined downregulate stage-specific genes (with log2(foldChange) <−1).

### 2.10. Protein–Protein Interaction Assays by Yeast Two-Hybrid Screening

Yeast two-hybrid experiments were performed using the Matchmaker Gold Yeast Two-Hybrid System (Takara). The coding sequences were first cloned into the entry vector pDONR/Zeo (ThermoFisher), and the correct ORF sequences were verified by Sanger sequencing. To this end, LDPGs were PCR-amplified with Gateway-compatible primers (Appendix A) using cDNAs of pooled male individuals as template. In the case of secreted proteins, the amplified ORF lacked the signal peptide. Then, the cloned ORFs were reamplified by a two step-PCR protocol allowing the creation of a recombination cassette containing the ORF flanked by 40-nucleotide tails homologous to the ends of the pGBKT7 bait vector at the cloning site. Linearized bait vectors and ORF cassettes were co-transformed into the Y2HGold yeast strain, and ORF cloning was performed by homologous recombination directly in yeast. Y2H screening for potential interacting partners of the bait proteins was performed via two methods: first, by directly testing pairs of candidates, and second, by testing the candidates as baits against a cDNA library obtained from a pool of total mRNAs from 100 *O. nana* male individuals cloned into pGAD-AD prey vectors.

Before screening, the self-activity of each bait clone was tested by mating with the Y187 strain harboring an empty pGADT7-AD vector, and then plating on SD/-His/-Leu/-Trp medium supplemented with 0, 1, 3, 5, or 10 mM 3-amino 1,2,4-triazole (3-AT). Each bait clone was then mated with the prey library containing approximately 4 × 106 individual clones and plated on low-stringency agar plates (SD/-Trp/-Leu/-His) supplemented with the optimal concentration of 3-AT based on the results of the self-activity test. To decrease the false-positive rate, after five days of growth at 30 °C, isolated colonies were spotted on high-stringency agar plates (SD/-Leu/-Trp/-Ade/-His) supplemented with 3-AT and allowed to grow another five days. Colony PCR on positive clones growing on high stringency medium was performed with primers flanking the cDNA insert on the pGAD-AD vector, and PCR products were directly Sanger-sequenced.

## 3. Results

### 3.1. Female-Biased Sex Ratio of Oithona nana

Between 2002 and 2017, 186 samples were collected in the Toulon Little Bay (Figure 1a), from which *O. nana* female and male adults were isolated (Figure 1b). Across fifteen years of observation, we noted minimum male/female ratios in February (0.11), maxima in September, October, and November (0.17), and a mean sex-ratio of 0.15 ± 0.11 over all years (Figure 1c). This monitoring of the sex-ratio showed a strong bias toward females, with relative stability over the years (ANOVA, *p* = 0.87).

### 3.2. Central Nervous System Labelling by Immunofluorescence

The central nervous system labeling with β3-tubulin in *O. nana* (Figure 2a,b) showed the male-specific presence of a lateral ganglion with a high density of β3-tubulin in the anterior part of the ganglion and a higher density of nuclei in its posterior part. We also observed the presence of β3-tubulin-rich post-ganglionic nerves that possibly connect the anterior part of the lateral ganglion to the tritocerebrum and/or the subesophageal ganglion. Not all males presented this labeling, and certain males contained one ganglion symmetrically on each side. However, no females presented this ganglion (Figure 2c). The α-tubulin labeling showed about seven to nine parallel afferent nerve fibers on the lateral part of the ganglion (Figure 2d,e) connected to free nerve endings located in the ventral part of the ganglion and in the external environment. Such labeling was absent in *O. nana* females. Together, these assays indicate the presence of a new nerve ganglion present only in *O. nana* males, located on the anterolateral part of the prosome.

### 3.3. Transcriptomic Support for Oithona nana Male Homogamety

To identify the most likely cause of this sex-ratio bias between potential environmental sex determination (ESD) and higher male mortality, we used SD-pop on four individual transcriptomes of both sexes to first determine the *O. nana* sexual system. Three sexual models were tested (no sex chromosomes, XY and ZW) with SD-pop; we obtained Bayesian information criteria (BIC) of 4,680,685.91, 4,680,725.39, and 4,680,670.66 for the no-sex-chromosomes, XY, and ZW models, respectively. Thus, the ZW model was preferred according to lowest BIC. This result is unlikely to be due to chance, as there were no runs on the 69 datasets for which the sex was permuted in which the ZW model had the lowest BIC. Eleven genes had a probability of being sex-linked in *O. nana* greater than 0.8; however, none of the SNPs in these genes showed the typical pattern of a fixed ZW SNP [21]. The four females genotyped were heterozygous, and the four males were homozygous (except for some SNPs, for which one male individual was not genotyped), indicating that the recombination suppression between the gametologs is recent, and that no or few mutations have been fixed independently in both gametolog copies. Annotation of these eleven genes shows that only one shared homology with other metazoan genes, that being ATP5H, which codes a subunit of the mitochondrial ATP synthase (Appendix A). As in Drosophila, *O. nana* ATP5H is encoded in the nucleus [43].

### 3.4. Lnr Domains Burst in the O. nana Proteome

To identify LDPs, we developed a HMM dedicated to *O. nana* LNR identification based on 31 conserved amino acid residues. In the *O. nana* proteome, 178 LNR and LNR-like domains were detected, encoded by 75 LDPGs, while a maximum of eight domains coded by six LDPGs were detected in four other copepods (Figure 3a,b). Among the 178 *O. nana* domains, 22 were canonical LNRs and 156 were LNR-like domains (Figure 3c). By comparing the structure of Notch, LNR, and LNR-like domains, we observed the loss of two cysteines (Figure 3c) in the LNR-like domains. Among the 75 LDPs, we identified nine different protein structure patterns (Figure 3d), including notably 47 LNR-only proteins, 12 trypsin-associated LDPs, and eight metallopeptidase-associated LDPs. Overall, LDPs were predicted to contain a maximum of five LNR domains and 13 LNR-like domains.

Forty-nine LDPs were predicted to be secreted (eLDP), six membranous (mLDPs), and twenty intracellular (iLDPs) (Appendix A). Among the iLDPs, two were associated with proteolytic domains, three associated with sugar–protein or protein–protein interaction domains (PAN/Appel, lectin, and ankyrin domains), and 13 (65%) were LNR-only proteins. Among the eLDPs, 18 (37%) contained proteolytic domains corresponding to a significant enrichment of proteolysis in eLDPs (hypergeometric test, *p* = 2.13 × $10^{-17}$); other eLDPs corresponded to LNR-only proteins (63%). The mLDPs were represented by one Notch protein, two proteins with LNR domains associated with lectin or thrombospondin domains, respectively, and three LNR-only proteins.

In phylogenetic trees based on nucleic acid sequences of the LNR and LNR-like domains (Figure 3e), only 17% of the nodes had support over 90%. Twenty-seven branch splits corresponded to tandem duplications involving 15 LDPGs, including Notch and a cluster of five trypsin-associated LDPGs coding three eLDPs and two iLDPs.

### 3.5. Oithona nana Male Gene Expression

Among the 15,399 genes predicted in the *O. nana* reference genome, 1233 (8%) were significantly differentially expressed in at least one of the five developmental stages. Among them, 619 genes were specifically upregulated in one stage, with 53 genes upregulated in the egg, 19 in nauplii, 75 in copepodids, 27 in adult females, and 445 in adult males (Figure 4b). The male-upregulated genes were categorized based on their functional annotation (Figure 4b).

#### 3.5.1. Upregulation of Lnr-Coding and Proteolytic Genes in Males

The 1233 differentially expressed genes contained 27 LDPGs (36% of total LDPGs) (Figure 4c). Of these 27 genes, 18 were specifically upregulated in adult males, producing a significant and robust enrichment of LDPGs in the adult male transcriptomes (fold change > 8; *p* = 2.95 × $10^{-12}$) (Figure 4c). Among the 445 male-specific genes, 27 were predicted to play a role in proteolysis, including 16 trypsins with three trypsin-associated LDPGs, showing significant enrichment of trypsin-coding genes in males (*p* = 1.73 × $10^{-5}$), as well as three metalloproteinases and five proteases inhibitors.

#### 3.5.2. Upregulation of Nervous-System-Associated Genes in Adult Males

Forty-eight upregulated genes in males had predicted functions in the nervous system (Appendix A). These included 36 genes related to neuropeptides and hormones, either through their metabolism (10 genes, seven of which encode enzymes involved in neuropeptide maturation and one of which is an allatostatin), through their transport and release (9 genes), or through neuropeptide or hormone receptors (17 genes, seven of which are FMRF amide receptors). Six genes were predicted to be involved in neuron polarization, four in the organization and growth guidance of axons and dendrites (including homologs to B4GAT1 and zig-like genes), one in the development and maintenance of sensory and motor neurons (IMPL2), and one in synapse formation (SYG-2, futsch-like).

#### 3.5.3. Upregulation of Amino-Acid Conversion into Neurotransmitters in Male Adults

We observed 10 upregulated genes in males predicted to play a role in amino acid metabolism (Figure 4d). This includes five enzymes that directly convert lysine, tyrosine, and glutamine into glutamate through the activity of one α-aminoadipic semialdehyde synthase (AASS), one tyrosine aminotransferase (TyrAT), and three glutaminases, respectively. Three other upregulated enzymes play a role in the formation of pyruvate: one alanine dehydrogenase (AlaDH), one serine dehydrogenase (SDH), and, indirectly, one phosphoglycerate mutase (PGM). Furthermore, two other upregulated enzymes are involved in the formation of glycine, one sarcosine dehydrogenase (SARDH), and one betaine-homocysteine methyltransferase (BHMT) (Figure 4d).

#### 3.5.4. Downregulation of Food Uptake Regulation in Male Adults

Three genes with predicted functions in food uptake regulation showed specific patterns in males. These included an increase of the mRNA encoding allatostatin, a neuropeptide known in arthropods to reduce food uptake, but also three male under-expressed genes: a crustacean cardioactive peptide (CCAP, a neuropeptide that triggers digestive enzymes activation), and the two bursicon protein subunits, which encode hormones known to be involved in intestinal and metabolic homeostasis.

### 3.6. Protein–Protein Interaction Network Involving Ldps and Igfbp

In order to further characterize the role of LDPs, we studied their potential protein interactions using a yeast two-hybrid system (Y2H). To this end, we selected 11 genes: seven male-overexpressed LDPGs, and four potential IGFBPs. The choice of the IGFBPs was made with the hypothesis that potential insulin-like androgenic gland hormone partners would be found in decapods [44]. We performed Y2H analysis using two different approaches (Appendix A): the first was a matrix-based screen with pairwise interaction assays, and the second aimed to identify potential interactors in the entire *O. nana* proteome by a random library screen. This latter screening approach was more time-consuming and was applied to only a subset of four genes (two LDPGs and two IGFBPs) used as bait proteins against a Y2H library constructed from *O. nana* cDNAs. Together, these two approaches allowed the reconstruction of a protein network containing 17 proteins, including two LDPs and one IGFBP used as baits (Figure 5a, Appendix A) and 14 interacting partners, of which six have orthologs in other metazoans and five have no orthologs, but at least one of which was detected using the InterProScan domain (Figure 5b).

On_LDP1, a putative extracellular trypsin-containing LDP, was found to form a homodimer and interact with a trypsin, two extracellular matrix proteins, and also an insulin-like growth factor binding protein (On_IGFBP7) that contained a trypsin inhibitor kazal domain. Based on its phylogeny, this protein is homologous to IGFBP7, also present in vertebrates (Appendix A). On_IGFBP7 formed a homodimer and interacted with three other proteins: one spondin-1 like protein (On_Spon1-like) containing a kazal domain, one thrombospondin domain-containing protein, and one vitellogenin 2-like protein (On_Vtg2). On_LDP2, coded by a gene upregulated in males (Figure 5c), interacted with nine proteins: one vitellogenin 2-like protein, three uncharacterized proteins, one homolog to somatomedin-B thrombospondin type 1 domain-containing protein, one homolog to the neuroendocrine protein 7b2 that contains a secretogranin V-like domain, one wnt5-like protein, one laminin 1 subunit β, and one furin-like 1 protein. No PPIs with IGF were detected, and no homologs to the decapod insulin-like androgenic gland hormone were found in the *O. nana* proteome.

## 4. Discussion

### 4.1. Zw Sexual System and High Mortality Rate of O. nana Males

Over 15 years of sampling, we observed a stable and strongly female-biased sex-ratio (1:9) in the Toulon Little Bay. A similar observation on a smaller time scale was previously made in another *O. nana* population [15] and in 132 other Oithonidae populations [17]. Two main factors could lead to this sex-ratio: a higher mortality of males or environmentally-induced sex determination. As we showed in our study, *O. nana* seems to have recently evolved genetic sex determination of the ZW type; thus, a 1:1 sex-ratio is expected in eggs, in the absence of sex-specific mortality in eggs. Therefore, a higher mortality rate of males is more likely to explain our in situ observations of the sex ratio. Moreover, these results are in accordance with the previously described risky behavior of Oithona males, that is, frequent motion to find females, thus making them more vulnerable to predators than immobile females [16].

### 4.2. The Development of a Male-Specific Nervous Ganglion Is Supported by Nervous System-Gene Expression

Immunofluorescence labeling of the *O. nana* nervous system demonstrated the existence of a ganglion in the anterolateral part of the male prosome. α-tubulin labeling showed nervous termination in the external environment, which suggests a sensory role to external cues. These observations constitute a new nervous and anatomical dimorphism between *O. nana* males and females [45,46]. The male-specific overexpression of numerous genes involved in the development of the nervous system supports the development of this male-specific ganglion at the molecular level, including axon guidance and synapse formation. From the upregulation of neuronal developmental genes in adult males, especially syg-2 and zig-8 normally expressed during the larval phase [47], we may infer ongoing formation of new axons and/or dendrites and synapses associated with the male lateral ganglion (Figure 6). On_LDPG2, a male-overexpressed gene under natural selection in the Mediterranean Sea populations has been shown to code an eLDP. The latter interacts with two proteins involved in the nervous system development, On_Wnt5 and On_Lamβ1 (Figure 6), notably important in axon guidance [48,49]. Therefore, through its protein–protein interactions, On_LDP2 may modulate neurogenesis in males and participate in the sexual dimorphism of the *O. nana* nervous system. It has been demonstrated that *O. davisae* males have a preference for virgins [28]; we could also speculate that the lateral ganglion may help in virgin-sensing of *O. nana* through the recognition of virgin-specific chemical cues.

### 4.3. Ldp-Driven Proteolysis Is Potentially Linked to the Male Ganglion Formation

The explosion of LDPGs in the *O. nana* genome is unique in metazoans and is associated with the formation of new protein structures containing proteolytic domains. Owing to the small size of the LNR domain (40 amino acids) and the substantial polymorphism within LNR domain sequences, the deep branches of the tree are weakly supported, and the evolutionary scenario resulting in the amplification of the domain remains undetermined. Two previous studies on *O. nana* population genomics [9,19] identified five LDPGs under natural selection, with point mutations within an LNR domain. These results reinforce the idea of an ongoing evolution of these domains, especially in *O. nana* males, and support an important role for these genes in the male biology. In metazoans, LNR domains are known to be involved in extracellular PPI [50] and cleavage-site-accessibility modulation [51]. Half of the eLDPs contain LNR domains only, and the other half associates with peptidases (trypsin and metalloproteinase). From the PPI network, we showed that On_LDP1 interacts with different types of extracellular proteins involved notably in tissue structure, energy storage, and extracellular proteolysis. On the other hand, transcriptomic analyses showed an enrichment of extracellular proteases in adult males (Figure 4d). This information supports the lysis of extracellular proteins in males involving complex protein interactions between eLDPs and trypsins. From our transcriptomic results on the expression of LDPs and nervous system genes, we hypothesize that the LDPs may play a role in the development of the male-specific ganglion by guiding proteolysis of the extracellular matrix around the ganglion. This lysis may help and guide the development of new neurites and the formation of synapses by preserving other anatomical structures. However, our study lacks a comprehensive or precise view of the function of the LDPs, due to the labor and time needed to optimize protocols for functional analysis on non-model species. Indeed, several RNAi assays were performed during this study without success due to the death of transfected individuals. However further investigations and methodological developments may help to validate our hypothesis of the role of LDPs in the neural development of *O. nana* male-specific lateral ganglion.

## 5. Conclusions

The hyper-motility of *O. nana* ZZ males and their faculty to find females could be one of the factors of the ecological success of the species and may explain the observed female-dominated sex ratio over a fifteen year course in the Toulon Little Bay. The male-specific nervous ganglion could play a role in its sexual behavior, its development was linked at the transcriptomic and protein levels to LDPGs, which seem to play an important role in the male-specific neurogenesis and proteolysis. The presence of LDPGs in other Oithona species should be investigated to better understand the evolution and role of these genes in other cyclopoid copepod species.

## Figures and Tables

**Figure 1 biology-10-00657-f001:**
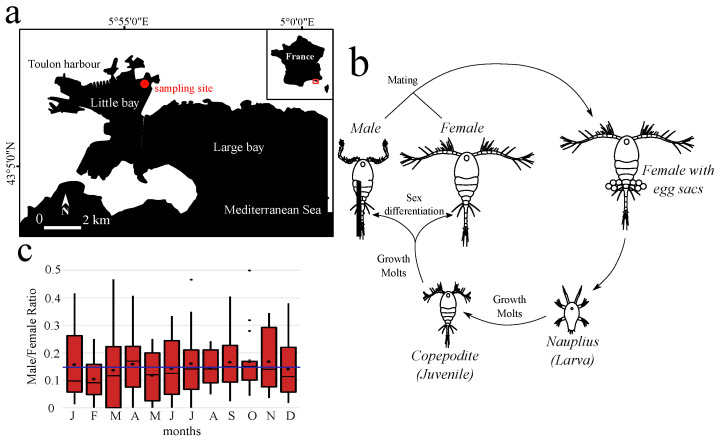
Life cycle and sex ratio of the copepod *Oithona nana* in the Toulon Little Bay. (**a**) Map of the sampling site in the Toulon Little Bay created with the “maps” R package 3.3.0 and modified with Inkscape 0.92.4. (**b**) The life cycle of *O. nana*. (**c**) Sex ratio of *O. nana* from 2002 to 2017 in the Toulon Little Bay. Black circles represent the monthly mean, the blue line represents the 15-year mean (0.15).

**Figure 2 biology-10-00657-f002:**
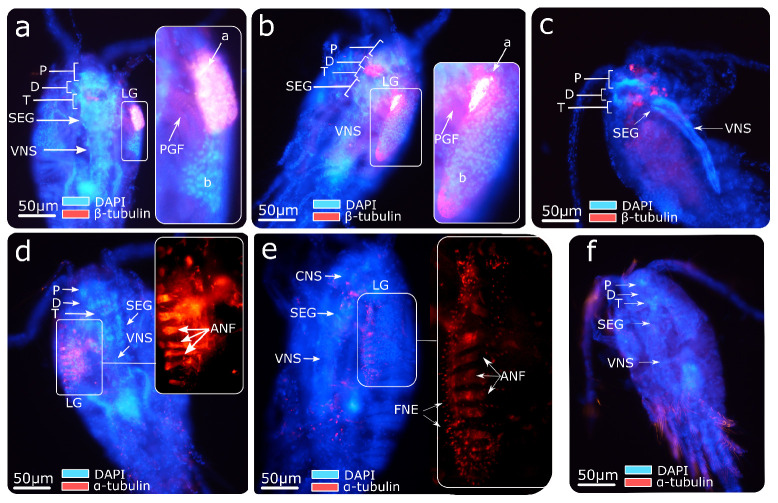
Immunofluorescence of the *Oithona nana* nervous system. (**a**) Male labeled with β3-tubulin and DAPI. (**b**) Male labeled with β3-tubulin and DAPI. (**c**) Female labeled with β3-tubulin and DAPI. (**d**) Male labeled with α-tubulin and DAPI. (**e**) Male labeled with α-tubulin and DAPI. (**f**) Female labeled with α-tubulin and DAPI. P: protocerebrum. D: deutocerebrum. T: tritocerebrum. SEG: suboesophageal ganglion. VNS: ventral nervous system. LG: lateral ganglion. PGF: post-ganglion fibers. a: β3-tubulin-rich area. b: nuclei-rich area. ANF: afferent nerve fibers. FNE: free nerve ending.

**Figure 3 biology-10-00657-f003:**
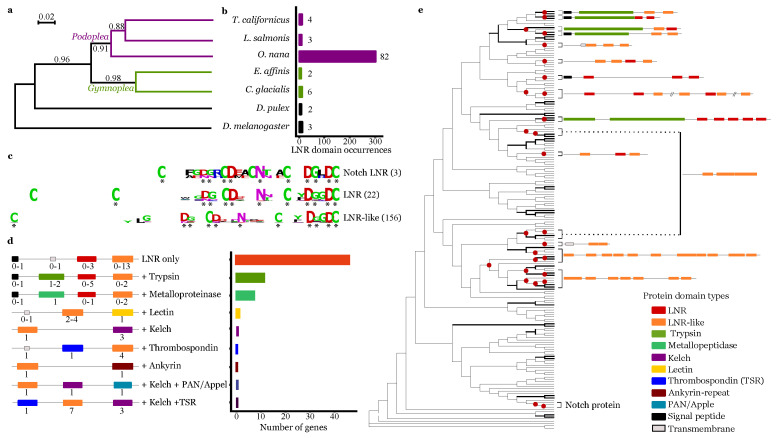
Lin-12 Notch Repeat (LNR) protein domain burst and domain associations in the *Oithona nana* proteome. (**a**) Phylogeny of five copepod species and two other arthropod species based on 18S ribosomal sequences. The numbers at internal branches show aLRT branch support. The scale bar represents the nucleotide substitution rate. (**b**) LNR domain occurrences in seven Arthropoda proteomes, detected by HMM. The number at the front of each bar corresponds to the number of detected genes. (**c**) Consensus sequences of the *O. nana* Notch-LNR, LNR, and LNR-like domains generated by WebLogo. The asterisks represent the conserved sites. (**d**) Schemata of the *O. nana* LNR and LNR-like protein structure. Numbers under each domain represent the possible occurrence range. The barplot represents the occurrence of the nine structures. (**e**) Phylogenetic tree of the *O. nana* LNR and LNR-like domains. Bold branches have aLRT support ≥ 0.90. The red circles represent tandem duplication.

**Figure 4 biology-10-00657-f004:**
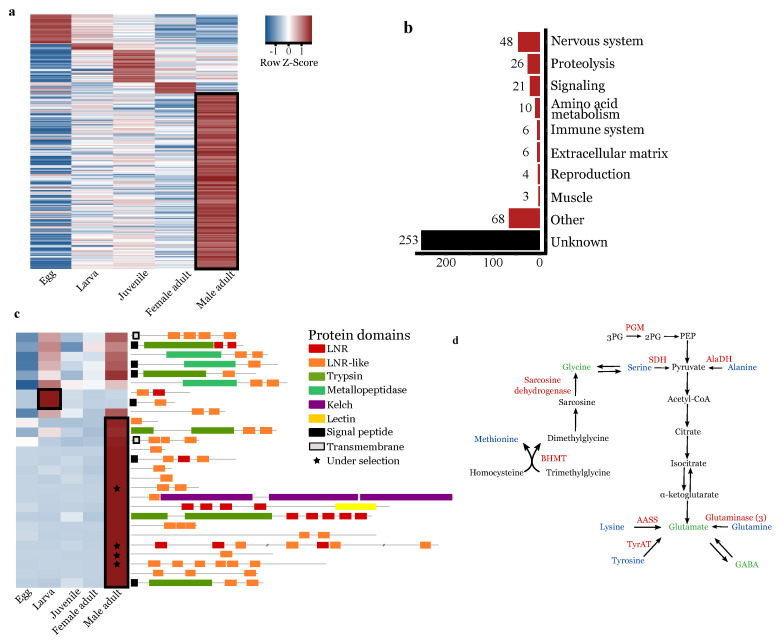
Differential expression analysis of the *Oithona nana* transcriptome. (**a**) Heatmap of the 1233 significantly differentially expressed genes in at least one of the five developmental stages. (**b**) Functional annotation distribution of 445 genes explicitly overexpressed in male adults. (**c**) Heatmap of the 27 significantly differentially expressed LDPGs and the composition of their protein domains. (**d**) Amino acid conversion to neurotransmitters in *O. nana* males. Overexpressed enzymes in males are indicated in red, amino acids in blue, and neurotransmitter amino acids in green.

**Figure 5 biology-10-00657-f005:**
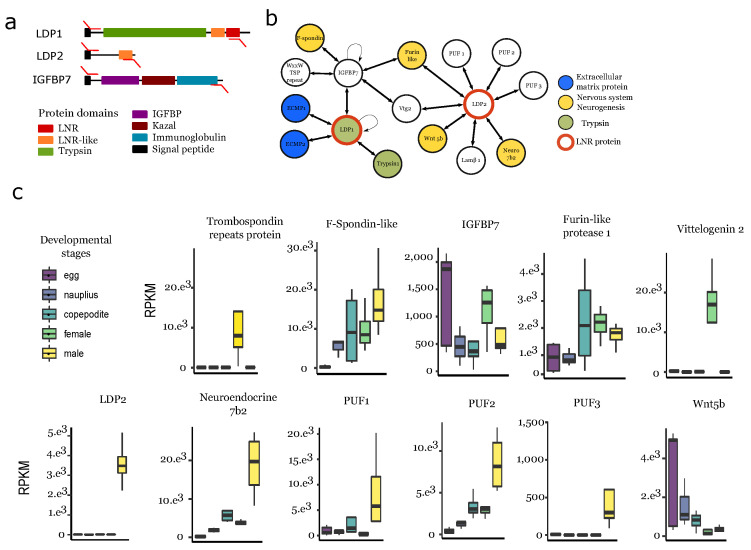
Protein–protein interaction of LNR-containing proteins in the *O. nana* male proteome. (**a**) Structure and expression of the PPI candidates. The red arrows represent PCR primers. (**b**) PPI network of LDPGs obtained by yeast two-hybrid assays. Lamβ1: Laminin subunit beta 1. Vtg2: Vitellogenin 2. PUF: protein of unknown function. (**c**) RPKM-normalized expression in the five developmental stages.

**Figure 6 biology-10-00657-f006:**
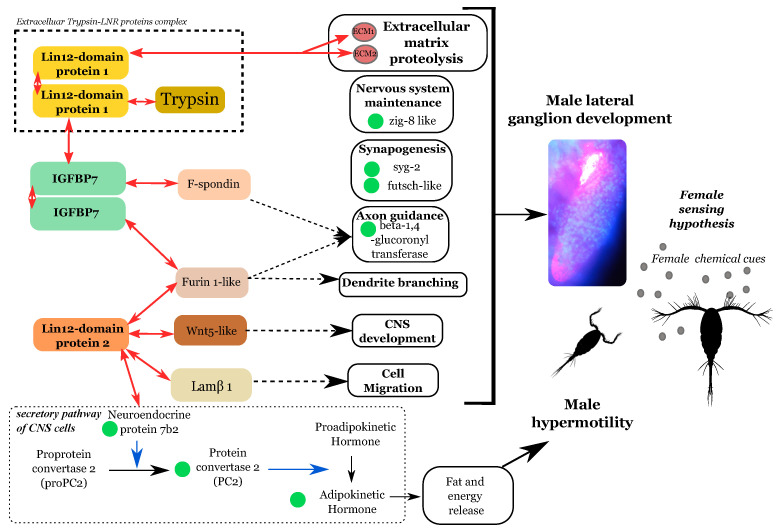
Hypothesis of the sensory role of the lateral ganglion in female search and the role of LDP-driven proteolysis. Red arrows represent protein–protein interactions found in this study. Blue arrows indicate enzymatic activities. Green circles represent overexpressed genes in males involved in the nervous system that were detected in this study. Dotted arrows indicate signaling pathways.

## Data Availability

The *O. nana* RNA-seq raw data are available at ENA (Appendix A).

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
