# Peer review of "Male Differentiation in the Marine Copepod Oithona nana Reveals the Development of a New Nervous Ganglion and Lin12-Notch-Repeat Protein-Associated Proteolysis"

_biology, 2021, doi:10.3390/biology10070657_

Round 1

Reviewer 1 Report

Siguer et al. investigate the possible molecular basis of O. nana sexual differentiation through a multi-approach study. The study includes immunofluorescence, genomics, transcriptomics, and protein-protein interaction from yeast two-hybrid experiments. Based on the authors' comprehensive analysis to support their hypothesis, I recommend accepting the manuscript based on the current presentation. There is only one minor suggestion to the present manuscript. In section 3.3, "Transcriptomic support for Oithona nana male homogamety," the result of the lowest Bayesian information criterion (BIC) to support the decision to prefer the ZW model is not present in the manuscript. However, the authors can provide this result as a figure in the supplementary material.

minor :

Siguer et al. investigate the possible molecular basis of O. nana sexual differentiation through a
multi-approach study. The study includes immunofluorescence, genomics, transcriptomics, and
protein-protein interaction from yeast two-hybrid experiments. Based on the authors'
comprehensive analysis to support their hypothesis, I recommend accepting the manuscript
based on the current presentation.
There is only one minor suggestion to the present manuscript. In section 3.3, "Transcriptomic
support for Oithona nana male homogamety," the result of the lowest Bayesian information
criterion (BIC) to support the decision to prefer the ZW model is not present in the manuscript.
However, the authors can provide this result as a figure in the supplementary material.

Author Response

Dear Reviewer,

We are very thankful for the reviewer comments; as recommended by the reviewer we added the BIC values on the manuscript Line 335 as follow:
“Three sexual models were tested (no sex chromosomes, XY and ZW) with SD-pop, we obtained Bayesian information criterions (BIC) of 4,680,685.91, 4,680,725.39, 4,680,670.66 for the no sex chromosomes, XY and ZW models, respectively. Thus, the ZW model was preferred according to lowest BIC.”

We think that this additional information will fit your expectations.

Reviewer 2 Report

Dear authors,

This manuscript is a follow-up of Oithona nana copepod from phenotype, behavior, and sex ratio to differential transcriptome and protein interaction analysis across a decade and a half to uncover morphological neural structures gene-linked to males. In general, the paper is well structured and presented and the experiments shown are rigorous. I only have one major question:

  I do not understand how the ZW model for sex determination is evidence of higher male mortality? In the results is mentioned that to find if the sex-bias ratio is due to environmentally-induced sex determination vs male mortality you performed a search for the best model and found it was the ZW; but I cannot see the association. Later it is discussed that a 1:1 sex ratio is expected in eggs, but no evidence is presented.

I also have some comments on a few ideas that I found unclear:

ln 239 I suggest a reference for the typical pattern of a fixed ZW SNP.
ln 280-284 Is confusing whether the 27 genes mentioned are the same or not in the latest mention of the pharagraph. I suggest rephrasing.
ln 378-381 LNR domain variability and evolution is unclear. I suggest rephrasing.

One final idea:

I would have liked to read more on the downregulated male genes and the possible food off-switch phenotype. Perhaps hypothesize a link between fasting and sexual-driven locomotion?

Best regads

Author Response

We are very thankful for the reviewer’s work which raises very interesting points out of our manuscript. The fact we predict a ZW sexual system in Oithona nana implies de facto a 1:1 sex ratio in the absence of sex specific mortality of the eggs. As well mentioned by the reviewer (although this remark is very legitimate), we did not verify the sex ratio in the eggs in this study for two reasons: we are not able to determine the sex of an egg using morphological features, and second the use of molecular data in eggs will need serious development considering the very low amount of DNA that can be retrieved from it (<<1ng).

Thus, In our study, we admit the higher mortality of male as previously described in Oithona davisae by Thomas Kiorboe in “Mate finding, mating, and population dynamics in a planktonic copepod Oithona davisae: There are too few males” which present very similar life history traits.

To clarify this important point, we clearly specify in the new version of the manuscript that the sex ratio in the eggs has not been measured (Line 357-359)

We are very thankful to the reviewer suggestion and take them into account for the ZW SNP pattern were we point out the SD-pop manuscript which provide the theoretical framework of SNP segregation in the different sexual systems. In this manuscript, we clearly describe multiple time the trade-off of fasting and sexual locomotion of the male. We also clearly illustrate the LDPG genes in the Figure 4.c and their different structures. Finally, we agree that the evolution of LNR is unclear, we steal add some new hypothesis but we cannot tell accurately the evolutionary history due to the shortness of the domain.

We hope that the modifications we made according to the reviewer comments will fit his expectations.

Reviewer 3 Report

The present study is an important and good contribution not only to the life history of a copepod, it will also help us to better understand the biology of an essential organism in the food chain. I personally really enjoyed the Figure 6 in the manuscript as it is going to help also general readers to understand how much ontogenetic development can influence important animals in the habitat.

Author Response

Dear Reviewer,

We would like to thank the reviewer for the great interest he shows on our scientific contribution.

Best regards,

The authors

Round 2

Reviewer 2 Report

The authors have addressed my concerns and suggestions. I have no further comments.